# Factors Affecting mHealth Technology Adoption in Developing Countries: The Case of Egypt

Ghada Refaat El Said 

Department of Management Information Systems, Future University in Egypt (FUE), 90th Street, Fifth Settlement, P.O. Box 11585, New Cairo 11835, Egypt; ghada.refaat@fue.edu.eg

**Abstract:** Mobile health apps are seeing rapid growth in the potential to improve access to healthcare services for disadvantaged communities, while enhancing the efficiency of the healthcare delivery value chain. Still, the adoption of mHealth apps is relatively low, especially in developing countries. In Egypt, an initiative for national-level healthcare coverage was launched in 2021, accompanied by a rise in mHealth start-ups. However, many of these projects did not progress beyond the pilot stage, with very little known about the antecedents of mHealth adoption for the Egyptian user. Semi-structured interviews were conducted with 22 Egyptians, aiming to uncover factors affecting the use of mHealth apps for Egyptian citizens. Some of these factors were introduced by previous studies, such as *Perceived Service Quality*, *Perceived Risk*, *Perceived Ease of Use*, and *Trust*. Others were not well established in the mHealth research strand, such as *Perceived Reputation* and *Perceived Familiarity*, while *Governance*, *Personalized Experience*, *Explain-ability*, *Interaction*, **Language**, and *Cultural Issues*, are novel factors introduced by the current research. The effect of these suggested independent variables on the willingness to adopt mHealth apps was validated using a survey administered to 150 Egyptians, confirming the significant positive effect of most of these factors on mHealth adoption in Egypt. This research contributes to methodology by introducing novel constructs in the mHealth research context, which might be specific to the target developing country. Practical implications were suggested for designers and healthcare service providers might increase the adoption of their apps in developing countries, such as Egypt.

**Keywords:** mHealth application adoption; trust; e-services; technology adoption in developing countries

## 1. Introduction

Mobile health (mHealth) is the use of mobile technologies for providing and supporting healthcare services. The global mHealth market size reached USD 50.7 billion in 2021 and is expected to expand with a growth rate of 11% from 2022 to [1]. mHealth technologies include mobile phone apps, platforms, and wearable devices, and it is expected that the mHealth apps segment will dominate the market with the largest revenue share of 75.4% in 2022 [1] mHealth apps are generally designed to provide healthcare information to the public, collect health data, monitor patients remotely, access medical records, conduct diagnoses, and assist in disease prevention and management. They are classified as education and teaching apps for healthcare workers, health and fitness apps for patients and the public, remote symptom-tracking apps, and others. The COVID-19 pandemic has led to the development of information apps, providing the public with information on the health measures to be taken and suggesting medical service providers [2]. mHealth apps are seeing rapid growth with the ever-growing smartphone user base. It gained traction during the COVID-19 pandemic due to infection fear and shutdown scenarios across the globe [2]. mHealth technology projects can assist in patient follow-up, healthcare workers' training and evaluation, supply chain and stock management, patients' education and awareness, disease surveillance, data collection/transfer, and more [3]. In the context of this research

scope, mHealth focuses on the use of mobile apps to serve as a search engine for users to suggest medical services.

Despite its convenience, the adoption of such e-services is still relatively low worldwide [4,5]. Such a low adoption rate hinders the optimization of medical resource distribution. It is crucial to explore the factors affecting the adoption of mHealth apps and investigate features to increase users' acceptance and usage. On the other hand, the potential benefits of mobile applications for healthcare purposes are particularly high for developing societies [6]. mHealth sustains cost-and-time-effective services to all stakeholders in the healthcare delivery value chain [7], ensuring affordable data collection [3] and extending access to healthcare services for patients in resource-poor and rural areas [8]. These cost-effective benefits, combined with the increased penetration of mobile phones in developing countries, have led to high investments in mHealth in these countries [9]. However, little is known about critical issues affecting the adoption of mHealth by patients in the developing world [10]. Despite the vast evidence in the literature highlighting the benefits of mHealth apps, poor public adoption remains a challenge [11]. Different patterns of adoption were found among different demographics, where socioeconomic factors were found to be one barrier to adoption [12]. Hence, more studies on mHealth app adoption are needed, especially for citizens from unexplored cultures. This study aims to provide implications for mHealth designers and service providers on how to promote the adoption of their applications among citizens in developing countries. In Egypt, there are serious barriers to e-services adoption that need to be overcome, such as a lack of trust, high costs, and weak network infrastructure [13]. Egypt has the largest population of internet and mobile users in the Middle East and North Africa region, with 59 million internet users, representing 52% of the total population, and 95.75 million mobile connections, representing 92.7% of the total population [14]. In 2021, the Egyptian government launched an initiative for national-level healthcare coverage by 2023, accompanied by governmental support for private providers of healthcare and mHealth start-ups. The rise of about 100 healthcare start-ups in Egypt was reported in 2021 [15]. Nevertheless, mHealth is still in its early stage in the Egyptian market and explanations of low mHealth penetration within this fast-growing mobile population have not been empirically verified.

This research looks at understanding the factors that affect the willingness to use mHealth applications in Egypt. It aims to provide implications for mHealth designers and service providers on how to promote the adoption of their applications in developing countries, such as Egypt. The paper starts with a literature review on mHealth adoption in developing countries, reported in Section 2. In Section 3, the research methodology is described, including an exploratory study of 22 semi-structured interviews to generate mature hypotheses. This was followed by a confirmatory e-survey administered to 150 Egyptian mobile app users. In Section 4, the data analysis and results are presented. In Section 5, a discussion of the findings is given, including the key theoretical and practical implications. Section 6 presents limitations and future research directions.

## 2. Literature Review
### 2.1. Previous Work on mHealth Adoption in Developing Countries

While developing communities are seeing rapid growth of internet and mobile app penetration, e-services adoption remains a major concern for researchers and service providers, especially in new application contexts such as healthcare [9]. Hence, several studies investigated factors affecting mHealth adoption in technologically developing communities from both sides: the healthcare workers and the users (patients).

An early study looking at mHealth use in Egypt collected survey data from 1014 participants across all Egyptian governorates [13]. The study reported a low percentage of Egyptian users (34% of the study sample size) had previous experience with mHealth apps. The study also reported negative attitudes of Egyptian users, associated with a low perception of trust, lack of user-friendliness, lack of infrastructure, and high perception of risk towards mHealth apps. In 2021, A survey was conducted [16] on mHealth users in the United

Arab Emirates and reported many barriers that needed to be overcome to have a regional impact, such as users' trust, privacy issues, high costs, and network infrastructure. Another study [17] developed an mHealth usage model in Bangladesh, which was empirically tested using data collected via a survey of 350 participants. Perceived Service Quality and Perceived Trust were reported to have significant explanatory power of intention to use mHealth. A similar study in Bangladesh suggested trust, service price, social influence, and facilitating conditions as key factors affecting mHealth adoption [18]. An adoption model of the mHealth application in Indonesia was developed by [6] based on 787 respondents. The model suggests that trust, facilitating conditions, and performance expectancy are important drivers of intention to use mHealth apps. An mHealth adoption model in China was suggested based on an extended Unified Theory of Acceptance and Use of Technology (UTAUT) [5]. The suggested model was validated using a survey completed by 746 patients. Performance Expectancy, Social Influence, and Trust in the application were suggested to have the strongest total effects on behavioral intention to use mHealth apps, followed by Privacy Risk and Facilitating Conditions. Also, in China, two similar studies by [18] and [19] applied the extended TAM, trust theory, and perceived risk theory to investigate the adoption of mHealth services. They concluded that Trust, Perceived Usefulness, and Perceived Ease of Use positively correlate with adoption, while Privacy and Risks negatively correlate with trust. In South Korea, a study [20] explored the intention of patients over the age of 40 to use mHealth Applications. They suggested that convenience, usefulness, and ease of use are key factors affecting mHealth adoption in this region.

While various models investigated challenges to systems' adoption in general [21,22], little work focused on mobile health adoption challenges specifically. Popular mobile health and fitness applications were studies to examine the reasons underlying both app adoption and abandonment of these applications [23]. The study explored reasons why citizens downloaded health apps in the first place, what motivated continued usage, and what reasons prompted app abandonment. It was reported that the top reasons for maintaining the use of a mobile health app were perceived effectiveness, efficacy, and customization ability. Reasons for app abandonment, on the other hand, were reported as mistrust of the service provider, lack of desired features such as tracking and notifications, and lack of motivation. [12] looked at mHealth app adoption among students in Ghana. The study observed gender inequalities in the use of mHealth apps, where adoption by males was higher. Other demographic and socioeconomic factors were reported to play a major role in mHealth adoption decisions, such as sociodemographic characteristics, specifically ethnicity, class of respondents and high average monthly income. The study suggested that these could be sociocultural barriers to potential mHealth app innovation adoption across citizens in Ghana.

Recently, [9] developed a Healthcare Worker mHealth Adoption Impact Model (HmAIM) to serve as a tool for the assessment of healthcare workers' mHealth adoption in the developing world. The model was validated through a suggested Healthcare workers' mHealth Adoption Questionnaire, administered to 104 nurses and midwives in Ghana who are users of a maternal mHealth application. The model suggests the significant effect of several factors (reliable infrastructure, usefulness, ownership, staffing, and technical support) on the intention to adopt. [10] validated the same model with 585 users, where new constructs were added to the model, related to user characteristics, portability, literacy, and funding. A summary of the findings of these studies is presented in Table 1.

### 2.2. Future Role of Emerging Technologies in Health Services

Some recent research [24,25] highlighted the possible future potential of blockchain-based multi-robot collaboration in healthcare environments. Such studies provide recommendations for academic and industrial researchers to invest more effort into the blockchain, multi-robot collaboration, and smart IoT technologies toward a smart healthcare environment. The work of [25] highlighted the possible future impact of digital twins, blockchain, the Internet of Things (IoT), and Artificial Intelligence (AI) in driving a revolution in the

healthcare sector. They suggested a blockchain-based collaborative digital twins' framework for decentralized epidemic alerting to combat COVID-19 and any future pandemics. The suggested framework has the potential to decentralize epidemic alerting to combat disease outbreaks and to facilitate secure real-time data exchange and analysis across multiple participants in times of pandemics. The work of [25]) highlighted the challenges and opportunities of integrating blockchain and multi-robot to combat pandemics. They proposed a framework to increase the intelligence, decentralization, and autonomous operations of connected multi-robot collaboration in the blockchain network. They proposed specific applications for Blockchain-Empowered Multi-Robot, such as integration between smart IoT devices and multi-robot in a smart hospital environment.

**Table 1.** Sample of Previous Work on mHealth Use in Developing Countries.

| Study | Developing Country | Trust | Perceived Ease of Use | Perceived Risk | Perceived Privacy | Cost | Infrastructure | Perceived Service Quality | Social Influence | Perceived Usefulness | Convenience | Technical Support | User Characteristics | Portability | Literacy | Perceived Ownership |
|---|---|---|---|---|---|---|---|---|---|---|---|---|---|---|---|---|
| [9] | Ghana | | | | | | ✓ | | | ✓ | | ✓ | | | | ✓ |
| [10] | | | | | | ✓ | ✓ | | | ✓ | | ✓ | ✓ | ✓ | ✓ | ✓ |
| [6] | Indonesia | ✓ | | | | | ✓ | ✓ | | | | | | | | |
| [4] | Bangladesh | ✓ | | | | ✓ | ✓ | | ✓ | | | | | | | |
| [17] | | ✓ | | | | | | ✓ | | | | | | | | |
| [16] | United Arab Emirates | ✓ | | | ✓ | ✓ | ✓ | | | | | | | | | |
| [13] | Egypt | ✓ | ✓ | ✓ | | | ✓ | | | | | | | | | |
| [5] | | ✓ | | | | | ✓ | ✓ | ✓ | ✓ | | | | | | |
| [18] | China | ✓ | ✓ | ✓ | ✓ | | | | | | ✓ | | | | | |
| [19] | | ✓ | ✓ | ✓ | ✓ | | | | | | ✓ | | | | | |
| [20] | South Korea | ✓ | | | | | | | | ✓ | ✓ | | | | | |

### 2.3. mHealth Start-Ups in Egypt

The past couple of years witnessed a wave of mHealth apps in Egypt, a country with a population of 100 million, one-third of whom are in low-income households in need of access to affordable healthcare [26]. The country suffered during COVID-19 with budget and doctor shortages. Hence, few mHealth solutions were developed via instant messaging, mobile phone consultations, and mobile videoconferencing. However, no research was conducted to assess the adoption of these solutions. In 2021, Egypt announced plans to roll out country-wide healthcare coverage by 2023, an effort that creates a wealth of opportunities for private providers of healthcare and start-ups [15]. Government endorses such projects as a strategy to complement actions related to achieving the health-related Millennium Development Goals and now the Sustainable Development Goals [3]. However, many of these projects did not progress beyond the pilot stage. One major challenge currently facing mobile health technology initiatives in the country is the lack of a foundation for evidence-based research to show the scaleup of such start-ups.

This study aims to contribute to the mHealth research strand by looking to understand the spectrum of factors that impact the adoption of mHealth interventions by users in the developing world. The research investigates users from Egypt, a developing country that is unexplored in this research context.

## 3. Methodology

This research started with an exploratory phase, where qualitative data was collected to generate research hypotheses. This was followed by confirmatory research using a survey.

### 3.1. Hypotheses Raising Study

Twenty-two semi-structured interview sessions were conducted over Zoom sessions between the researcher and interviewees. **The Sample**: Convenience sampling was used and included 22 university students from different specializations. Gender was equally distributed, and the age range varied between 18 and 24. No compensation was provided to students. **Procedures**: All sessions were 30 min in average duration. The interviews started with a few guiding questions: *"What are the factors which would motivate you to use an m-health app?"* and *"What are the characteristics of the m-health app which would encourage you to use it?"*. These questions were followed by a series of reflective questions to elicit in-depth responses. **Thematic Analysis**: Interview answers were transcribed and analyzed through thematic analysis. Unlike textual analysis, thematic analysis goes beyond counting phrases or words and moves on to identifying patterns and themes across datasets, describing the phenomenon under investigation [27], and creating meaningful patterns to identify unfolded themes emergent from the data [28]. Identifying a theme does not necessarily yield the frequency at which a theme occurs. Ideally, the theme will occur numerous times across the dataset, but a higher frequency does not necessarily mean that the theme is more important to understanding the data. Based on the thematic analysis of the interview data, 13 themes were generated. A sample of interview statements, themes and sub-themes is presented in Table 2. Some research constructs were hypothesized to affect mHealth adoption for Egyptian users. Those constructs are:

| | | |
|---|---|---|
| Language | Culture issues | Perceived Reputation |
| Trust | Perceived Familiarity | Perceived Service Quality |
| Perceived Ease of Use | Perceived Risk | Governance |
| Personalized Experience | Portability | Explain-ability |
| Interactivity | | |

**Table 2.** Interview Thematic Analysis: Research Construct Identification.

| Sample of Answers Statements | Sub-Theme | Theme |
|---|---|---|
| *"The app needs to be in both **Arabic and English** . . . I need to be able to switch between these two languages smoothly"*<br>*"I expect to receive information in **medical terms** with explanation in **natural language**"* | - Native language<br>- Switch between languages<br>- Medical terms/natural language | Language |
| *"I need the option to select the **gender** of the healthcare provider, I do prefer to be in contact with a female doctor"*<br>*"If this app is expected to be used nation-wide, **illiterate users** should be interacting with voice or comprehensive icons"* | - Gender preference<br>- Illiteracy | Culture |
| *"I would **trust** the mobile app if the service provider were **well reputed** and **widely recognized**"*<br>*"The app would be more **trustworthy** if endorsed by a known expert"*<br>*" . . . . obtained **certification** to operate from a governmental official"* | - Reputable service provider<br>- Widely recognized<br>- Endorsement of known experts<br>- Certification of operation<br>- Trustworthy | Perceived Reputation and Trust |
| *"I prefer to use app I am **familiar** with."*<br>*"I prefer an app I **used before**, or an app designed like what I am **familiar** with, this will ease the process of searching"*<br>*"I would feel confident to use an app **used by people I know**"* | - Familiar design<br>- Previous Use<br>- Used by familiar people | Perceived Familiarity |
| *" . . . **accurate medical information** given"*<br>*"I should receive a **follow-up** communication, to check my feedback and health status."*<br>*"Allow users' **ratings and reviews** and they are **reliable**"* | - Accurate information<br>- Follow-up<br>- Reliable users' rating/reviews | Perceived Service Quality |

**Table 2.** *Cont.*

| Sample of Answers Statements | Sub-Theme | Theme |
|---|---|---|
| *"I need to find the interface **easy to use**"*<br>*"I would be motivated to use the app if the use instructions are **comprehensive**"*<br>*"Using the App does not **require external help/instructions**"* | - Easy to use interface<br>- Comprehensive instructions<br>- Need for external help | Perceived Ease of Use |
| *"Such mobile apps are not **secure**; my data would be **miss-used** when using such apps"*<br>*"Who will be **governing** the app? Private sector or the ministry of health?"*<br>*"**users' data security** is a key issue for using or not using the app"*<br>*"I need to be given a declaration concerning my health **data privacy and confidentiality**"* | - Data security<br>- Data miss-use<br>- Privacy/confidentiality terms<br>- Governing entity | *Perceived Risk and Governance* |
| *"The app should **recognize my preferences** based on previous use"*<br>*"User data would be saved in a profile, such as location and health insurances, based on which the **search results could be tailored**"*<br>*"The app should not suggest to me a healthcare provider, I already rated low"* | - Keeping user profile<br>- Tailored search results<br>- Recognition of user preferences | *Personalized Experience* |
| *"Will the app work on **different types of mobiles** and systems? What about **Internet connectivity instability** . . . ? Not all users have **up-to-date smart phone . . . . Connections are not reliable** in many areas" "I would prefer to use the app from my **laptop**"* | - Compatibility with Mobile types and other devices<br>- Internet connection variation | *Portability* |
| *"when recommending a healthcare service provider, the app should provide all **details of the expected service**, ex.: fees, waiting time, insurance coverage . . . "*<br>*"The mechanism of search engine should be **transparent** . . . what are the criteria based on which the search results are sorted?"*<br>*"The **full experience of user review** should be cited, not only ranking"* | - Comprehensive details of recommended medical service<br>- Explanation of selected criteria<br>- Reporting of detailed users reviews | *Explain-ability* |
| *"I understood that the interaction is asynchronous . . . .in critical health cases **real time communication is needed** . . . or in case of an **error in the app**"*<br>*"I do prefer to communicate with a **human** not an app especially when it comes to health consultation"* | - -Real-time health consultation<br>- -Real-time troubleshooting<br>- -Human intervention, if needed | *Interactivity* |

*3.2. Hypotheses Testing Study*

An e-survey was designed using most construct measuring items drawn from the literature, where they were all reported to be reliable and valid. A 5-point Likert scale was used. The first section of the survey included basic demographic information, and the second section included items measuring the research constructs. *The Sample:* Convenience sampling was used in this research, as participation in the study was voluntary. Participants were all Egyptians living in Cairo and from different proficiency and background (sales representatives, web designers, accountants, students, human resources staff, and teaching staff). The URL of the e-survey, composed on Google docs, was circulated by the researcher via WhatsApp groups, with an introductory note encouraging individuals to fill out the survey. The cover page of the e-survey included introductory notes about the research aim, its objectives, clarification that completing the survey is voluntary and anonymous, and that results will be used for research purposes only. In total, 150 valid responses were received. Table 3 illustrates the demographic characteristics of the sample.

**Table 3.** Demographic Characteristics of Sample.

| Item | Frequency | % | Item | Frequency | % |
|---|---|---|---|---|---|
| Gender | | | Use of Search Engines | | |
| Male | 68 | 45% | Never | 0 | 0% |
| Female | 82 | 55% | Rarely | 5 | 3% |
| | | | Occasionally | 27 | 18% |
| | | | Frequently | 118 | 79% |
| Age Range | | | Browsing the Net | | |
| <20 | 12 | 8% | Never | 0 | 0% |
| 20–29 | 21 | 14% | Rarely | 5 | 3% |
| 30–39 | 29 | 19% | Occasionally | 37 | 25% |
| 40–49 | 41 | 28% | Frequently | 108 | 72% |
| 50–60 | 32 | 21% | | | |
| >60 | 15 | 10% | | | |
| Technology Use | | | How long using the Net | | |
| Novice | 15 | 10% | ≤1 year | 0 | 0% |
| Intermediate | 42 | 28% | 2–3 years | 32 | 21% |
| Professional | 93 | 62% | 4–5 years | 41 | 27% |
| | | | ≥6 years | 77 | 52% |
| Previous Use of M-Health App | | | Language Preference | | |
| Never | 42 | 28% | Arabic | 57 | 38% |
| Rarely | 30 | 20% | English | 62 | 41% |
| Occasionally | 46 | 31% | Arabic & English | 31 | 21% |
| Frequently | 32 | 21% | | | |
| Previous Use of M-Health App to search and/or book medical service | | | Previous Use of M-Health App to get online medical service | | |
| Never | 42 | 28% | Never | 42 | 28% |
| Rarely | 30 | 20% | Rarely | 30 | 20% |
| Occasionally | 46 | 31% | Occasionally | 46 | 31% |
| Frequently | 32 | 21% | Frequently | 32 | 21% |

## 4. Analysis and Results

### 4.1. Demographic Characteristics

The sample's age range varies from <20 to >60 years old, with the majority having an intermediate or expert level in using the Internet for more than 2 to 6 years. Some 57% prefer to use an Arabic interface. While most participants (93%) consider themselves intermediate or expert users of technology, with frequent use of search engines (79% of the sample) and frequent use of web browsers (72% of the sample), 48% of the sample never or rarely had previously used m-health apps, while the remaining 52% occasionally or frequently used m-health app for receiving medical advice, search, and book for medical services. Gender is almost balanced: males represent 45% of the sample and females represent 55%. SPSS was used in the data analysis. Although most of the items used in this study were drawn from the literature, where they are reported to be reliable, nevertheless validity and reliability of the survey were checked as follows.

### 4.2. Construct Reliability

According to [29], a value of 0.80 or greater suggests evidence of strong composite reliability, and the Average Variance Extracted (AVE) should also be greater than 0.50 to demonstrate significant variance. Meanwhile, Cronbach's Alpha, measuring how well a set of items measures a single unidirectional latent construct, is suggested to be at least 0.6 [29]. As illustrated in Table 4, the constructs: Perceived Risk, Personalized Experience, Interactivity, and Culture had composite reliability, AVE, and Cronbach's Alpha less than 0.5. Accordingly, these constructs were dropped from the analysis.

**Table 4.** Construct Reliability.

| Construct | Composite Reliability | AVE | Cronbach Alpha | Construct | Composite Reliability | AVE | Cronbach Alpha |
|---|---|---|---|---|---|---|---|
| Perceived Reputation-PREP | 0.881 | 0.648 | 0.840 | Willingness to Use-WTU | 0.931 | 0.772 | 0.902 |
| Perceived Familiarity-PFAM | **0.401** | **0.418** | **0.449** | Governance-GOV | 0.839 | 0.726 | 0.605 |
| Perceived Service Quality-PSQ | 0.897 | 0.812 | 0.745 | Portability-POR | 0.891 | 0.804 | 0.801 |
| Perceived Risk-PRSK | **0.393** | **0.436** | **0.459** | Personalized Experience-PEREXP | **0.425** | **0.405** | **0.487** |
| Perceived Ease of Use-PEOU | 0.846 | 0.650 | 0.757 | Explain-ability-Exp | 0.869 | 0.790 | 0.618 |
| Interactivity-Inter | **0.301** | **0.402** | **0.339** | Language-LANG | 0.867 | 0.856 | 0.735 |
| Trust-TRST | 0.832 | 0.612 | 0.831 | Culture-CUL | **0.412** | **0.434** | **0.451** |

*4.3. Item Reliability*

According to [30], item loading and item-construct correlation should be at least 0.60. Based on the data in Table 5, items PREP1, FAM2, FAM3, PSQ3, PSQ4, PRSK2, TRST3, WTU3, EXPL2, POR1, PEXP1, and CUl1 failed to meet this criterion, and these items were dropped. All remaining items were found to achieve adequate reliability. Furthermore, all item-construct correlations exceeded 0.7. Accordingly, it could be concluded that the remaining items measuring all constructs had adequate reliability.

**Table 5.** Item Reliability.

| Construct/Source from Literature/Item | Item Loadings | Item Construct Correlation |
|---|---|---|
| **Perceived Reputation—PREP** [31] | | |
| PREP1—This app has a bad reputation | **0.3228** | **0.396** |
| PREP2—This app is well known | 0.7117 | 0.785 |
| PREP3—This app has a good reputation | 0.8931 | 0.931 |
| **Perceived Familiarity—PFAM** [32] | | |
| PFAM1—I am familiar with searching for information on this app | 0.8289 | 0.867 |
| PFAM2—I am familiar with paying for services on this app | **0.3128** | **0.429** |
| PFAM3—I am familiar with this app | **0.3572** | **0.340** |
| PFAM4—I am familiar with doctors' ratings on this app | 0.7509 | 0.777 |
| **Perceived Service Quality—PSQ** [33] | | |
| PSQ1—I would recommend this app to friends | 0.6012 | 0.787 |
| PSQ2—This app is reliable and accurate | 0.7356 | 0.789 |
| PSQ3—This app responds quickly to problems | **0.4678** | **0.545** |
| PSQ4—This app provides contact services for users | **0.5321** | **0.512** |
| **Perceived Risk—PRSK** [31] | | |
| PRSK1—There is too much uncertainty associated with this app | 0.8123 | 0.896 |
| PRSK2—Compared with other ways, online payment is risky | **0.5456** | **0.598** |
| PRSK3—There could be negative consequences of online payment | 0.8789 | 0.868 |
| **Perceived Ease of Use—PEOU** [34] | | |
| PPEOU1—Learning how to use this app is easy to use | 0.8215 | 0.825 |
| PEOU2—My interaction with this app is clear | 0.8538 | 0.882 |
| PPEOU3—I find this app easy to use | 0.8541 | 0.867 |

**Table 5.** *Cont.*

| Construct/Source from Literature/Item | Item Loadings | Item Construct Correlation |
|---|---|---|
| **Trust—TRST** [32] | | |
| TRST1—I trust this app is reliable | 0.8439 | 0.899 |
| TRST2—I believe that this app is trustworthy | 0.8742 | 0.904 |
| TRST3—I trust providing personal information to this app | **0.5433** | **0.504** |
| **Willingness to Use—WTU** [31] | | |
| WTU1—I am very likely to use this app | 0.7124 | 0.666 |
| WTU2—I am very likely to use this app in 3 months | 0.7926 | 0.865 |
| WTU3—I am very likely to use this app in the next year | **0.5533** | **0.571** |
| **Governance** | | |
| GOV1—It is important to know who is governing the app | 0.6542 | 0.698 |
| GOV2—It is important to know who is licensing the app to operate | 0.7931 | 0.878 |
| **Portability** | | |
| POR1—The app can be used on various devices | **0.5412** | **0.598** |
| POR2—The app can operate with weak internet connectivity | 0.8789 | 0.868 |
| **Personalized Experience** | | |
| PEREXP1—the app recognizes my profile once I log in | **0.4124** | **0.466** |
| PEREXP2—the app tailors search results based on my profile | 0.7931 | 0.858 |
| **Explain-ability** | | |
| EXPL1—the app provides details of recommended medical service | 0.8132 | 0.869 |
| EXPL2—the app provides an explanation of search criteria | **0.5431** | **0.578** |
| EXPL3—the app provides details of user reviews | 0.8788 | 0.869 |
| **Interactivity** | | |
| INTER1—the app provides real-time health consultation | 0.8412 | 0.898 |
| INTER2—the app provides real-time troubleshooting | 0.8789 | 0.868 |
| **Language** | | |
| LANG1—the app supports Arabic and English data entry | 0.8513 | 0.808 |
| LANG2—the app owns Arabic and English interfaces | 0.8712 | 0.834 |
| **Culture** | | |
| CUL1—The gender of the healthcare service provider matters to me | **0.3412** | **0.348** |
| CUL2—A specific level of literacy is needed to use the app | 0.6689 | 0.668 |

### 4.4. Item Correlations

Each item correlates more highly with other items measuring the same construct than with other items measuring other constructs, as evidence of discriminant validity [30]. Furthermore, each item's loading is much higher on its assigned construct than on the other constructs [30]. Furthermore, all items were found to have much higher loading in their assigned constructs than in the other constructs (see Appendix A). This suggests the discriminant validity of all the used items.

### 4.5. Construct Validity

Construct validity was measured using the correlation coefficient between each construct and its associated items using Pearson Correlation Coefficient. A significant correlation was found between all constructs and all their associate items, as presented in Appendix B.

### 4.6. Final Reliability

All the reliability coefficients satisfied the minimum Cronbach alpha not less than 0.80. as listed in Table 6. This suggested that the instrument was sufficiently reliable. In general, nine constructs and 19 items were identified, including three items to measure

Perceived Ease of Use and two items to measure each of the following constructs: Perceived Reputation, Perceived Familiarity, Perceived Service Quality, Trust, Willingness to Use, Governance, Explain-ability, and Language.

**Table 6.** Instrument Reliability.

| Construct | Number of Items | Cronbach Alpha |
|:---:|:---:|:---:|
| PREP | 2 | 0.821 |
| PFAM | 2 | 0.869 |
| PSQ | 2 | 0.801 |
| PEOU | 3 | 0.897 |
| TRST | 2 | 0.802 |
| WTU | 2 | 0.912 |
| GOV | 2 | 0.867 |
| EXPL | 2 | 0.923 |
| LANG | 2 | 0.834 |

*4.7. Regression*

Multiple linear regression was used to test if predictor constructs (Perceived Reputation, Perceived Familiarity, Perceived Service Quality, Perceived Ease of Use, Trust, Governance, Explain-ability, and Language) significantly predicted response construct (Willingness to Use).

Beta Coefficient was found significant ($p < 0.01$), confirming the importance of all constructs as predictors of the Willingness to Use, except for Perceived Familiarity. The regression includes $R^2 = 0.887$, expressing that 88% of the change in WTU is due to changes in the PREP, PEOU, GOV, LANG, EXPL, TRST, and PSQ, with the loading of 0.892, 0.887, 0.770, 0.722, 0.512, 0.511, 0.432 respectively. According to data reported in Table 7, Perceived Reputation, Perceived Ease of Use, Governance, and Language are the most important factors leading to Willingness to Use, followed by Explain-ability, Trust, and Perceived Service Quality.

**Table 7.** Linear Regression.

| Dependent Variable | R2 | Independent Variables | Coefficient (T-Value) | Significance |
|:---:|:---:|:---:|:---:|:---:|
| WTU | 0.887 | PREP | 0.892 (13.258 **) | 0.000 |
| | | PEOU | 0.887 (12.799 **) | 0.000 |
| | | GOV | 0.770 (11.899 **) | 0.000 |
| | | LANG | 0.722 (10.911 **) | 0.000 |
| | | EXPL | 0.512 (8.901 **) | 0.000 |
| | | TRST | 0.511 (8.999 **) | 0.000 |
| | | PSQ | 0.432 (6.799 **) | 0.000 |

** significant ($\rho < 0.01$).

## 5. Discussion

The potential benefits of mobile applications for healthcare purposes are particularly high for developing societies such as Egypt. However, despite its fast-growing mobile population, mHealth is still in its early stage in the Egyptian market. This research looked at understanding the factors that affect mHealth adoption in Egypt. Exploratory semi-structured interviews were conducted with 22 users. Several themes were identified as main factors affecting mHealth use for the selected sample, which are listed in Table 2

and were identified as research constructs. A survey was developed to measure these constructs and was administered to 150 Egyptian mobile app users. A number of these constructs were validated as having an impact on mHealth adoption within the Egyptian sample that participated in the study. Some of these reported themes are suggested, in other technology adoption contexts, such as e-commerce, to reflect local contextual or cultural factors. Some studies argue that Hofstede's UA-uncertainty avoidance [35]) culture variable plays an important role in the attitude forming toward technology adoption in developing countries [36], namely in the Arab countries [37]. These studies argued that people in a developing culture and with a high score for UA might not gratuitously accept new technologies, especially these which are directly related to their lives. Technology users with high uncertainty avoidance are expected to have a low tolerance for ambiguity and uncertainty while using mHealth apps can be seen as an example of an activity with an uncertain outcome [12]. More specifically, previous research suggested that for users from uncertainty avoidance culture, such as the Egyptians, the effect of Perceived Reputation [38], Perceived Familiarity [20], Perceived Risk [39], and Trust [33] on technology adoption are significant. Receiving health advice from a mobile application presents numerous risks for users [12]. The process becomes less uncertain when the app provider is one with a good reputation and is trustworthy; uncertainty might also decrease when the user is familiar with this app. It is, therefore, tempting to explain the influence of the mHealth app's Perceived Reputation, Familiarity, Risk, and Trust on adoption within the sample of the current research. While these factors might also be important for low uncertainty avoidance cultures such as the US and Australia, nevertheless, it is suggested that the effect of these factors on the adoption of mHealth apps is relatively more important in high uncertainty avoidance cultures.

## 6. Research Conclusions and Contributions

This research contributes to methodology by introducing novel constructs in the mHealth research context, which might be specific to the target region, a developing country, that is unexplored in the mHealth research context. The findings also provide practical implications for the ways in which mHealth designers and healthcare service providers might increase the adoption of their apps in developing countries, such as Egypt.

**Research Contributions to Knowledge:** Perceived Reputation was suggested by most interviewees (17 out of 22 interviewees) and validated by the survey (t = 0.892; *p* < 0.01) as having a significant positive impact on mHealth adoption. Interviewees expressed the importance of the reputation, recognition, and experts' endorsement of the mHealth service providers. Perceived Familiarity was also identified by a high majority of interviewees (14 out of 22 interviewees), although it failed to be validated by the survey. Previous use and familiar design were cited as encouraging factors to use mHealth. While Perceived Reputation and Perceived Familiarity are not well established in previous mHealth research in developing countries, these two factors are well established as adoption predictors for other e-services, such as e-payment (Lei et al., 2021; Yao et al., 2008) and e-education (Al-raimi et al., 2015). While in the e-commerce research strand, Perceived Reputation (Pavlou, 2003; De Ruyter et al., 2001) and Perceived Familiarity (Gefen et al., 2003; Bhattacherjee, 2002) are well established as the main factors of adoption. An important contribution of this research is the validation of these two antecedents of mHealth adoption for Egyptian users. Trust and Perceived Service Quality were suggested by half of the interviewees, while trust as a predictor of adoption was validated by the survey (t = 0.511; *p* <0.01), Perceived Service Quality was not. Being directly related to users' life and health, interviewees highlighted the fact that they need to be sure of the quality of healthcare services provided by the app (information accuracy, reliable rating, and reviews) and also need to trust the mHealth app and service provider. However, few linked these two perceptions with endorsement and approved certificates of operation for the app from governmental authorities. This finding supports previous research highlighting the importance of user trust in mHealth in other developing countries, such as in United Arab Emirates (Messeih, 2021), China

(Zhang et al., 2019; Meng et al., 2019), Indonesia (Octavius et al., 2021), and Bangladesh (Akter et al., 2013; Alam et al., 2020). The importance of Perceived Service Quality was also supported by previous work in the same context (Octavius et al., 2021; Zhang et al., 2019; Akter et al., 2013). Perceived Risk was highlighted by the number of interviewees (10 out of 22) in terms of fear of confidential data misuse, where some participants suggested that Governance is crucial in terms of governing entities to ensure data privacy and security. While both Perceived Risk and Governance failed to be validated by the survey, Perceived Risk was supported by some previous research on mHealth in other developing countries (Meng et al., 2019; Deng et al., 2018; Mansour, 2017). The highest percentage of interviewees (85%) identified Perceived Ease of Use, and the same percentage identified Explain-ability, in terms of the comprehensive details of recommended medical service and explanation of selected criteria, as the main features of mHealth leading to adoption, also validated by the survey, t = 0.887 and t = 0.512 respectively, as main predictors of mHealth adoption. This expected effect of ease of use on adoption is supported by various mHealth adoption models (Mansour, 2017; Meng et al., 2019; Deng et al., 2018; Lee et al., 2017).

Although all participants self-evaluated their English language level between Intermediate and Advanced, they still support bilingual (Arabic and English) support. Languages were suggested as a vital feature for mHealth use by 76% of interviewees, supported by the survey (t = 0.722; $p < 0.01$), where participants showed a preference to be able to switch between both languages and receive medical terms in their natural language. As most of the current study sample are familiar with mobile technology and have good English language skills, the effect of Language and other Cultural Issues (such as literacy and preference of healthcare provider gender) could be increased for users with diverse characteristics. It is worth mentioning that the illiterate rate in Egypt reached 24.6% in 2019 (CAPMAS, 2020). Other mHealth features were suggested by the current research, identified by a varying number of participants, such as Portability (60% of interviewees), Personalized Experience (55%), and Interactivity (51%). Participants showed preference to be able to use the mHealth with different types of mobile phones and under different conditions of Internet connectivity (portability), to be able to save user's profile and receive search results based on pre-set preferences (personalized experience), and to receive real-time troubleshooting human intervention if needed (Interactivity).

**Contributions to Practice:** Practical recommendations could be given to designers and service providers. According to the results, increased willingness to use mHealth applications could be achieved by increasing users' perception of reputation, which could be reached by describing the organization's history and policies for users' satisfaction, by listing previous users' previews and ratings, in addition to the presence of a governing entity to certify operation, ensure data confidentiality, and decrease the perceived level of risk. Developing familiarity can be achieved through regular advertising to increase the recognition of the app. Perceived Ease of Use can be achieved through interface consistency and obviousness, with a high level of explain-ability of medical terms in natural language, with personalized search results while providing real-time troubleshooting. Trust can be achieved by clarifying policies for payment refunds and secured online payment, which is also essential in dealing with users' perceptions of risk. For wider adoption of mHealth, some user characteristics would be considered, such as preference for the Arabic language, variation of mobile device types, and instability of internet connection in some rural areas in the country. Such findings can be a starting point to elaborate on to specify design guidelines for mobile health services targeting users from Egypt with its 100-million-strong domestic market.

## 7. Research Limitations and Future Work

Convenience sampling was employed in the research, where most of the sample are familiar with mobile technology, with good English language and located in the capital, Cairo, with the best mobile and internet infrastructure in the country. Convenience sampling was used in the current research for its cost and time-saving. It is a simple technique

that allows the researcher to obtain basic data and trends regarding the phenomena under investigation without the complications of using a randomized sample. Still, the sample could be a limitation in the current research. As most of the sample were familiar with mobile technology and with good English language, this might hinder a possible socio-cultural effect on adoption. Future studies should incorporate participants from rural areas with various demographic characteristics to assess regional differences and possible cultural dimensions.

**Funding:** This research received no external funding.

**Data Availability Statement:** All data used are included in this published article.

**Conflicts of Interest:** The authors declare no conflict of interest.

## Abbreviations

The following abbreviations are used in this manuscript:

| | |
|---|---|
| PREP | Perceived Reputation |
| PFAM | Perceived Familiarity |
| TRST | Trust |
| PSQ | Perceived Service Quality |
| PRSK | Perceived Risk |
| CUL | Culture |
| PEOU | Perceived Ease of Use |
| GOV | Governance |
| LANG | Language |
| WTU | Willingness to Use |
| PEREXP | Personalized Experience |
| EXP | Explain-ability |
| INTER | Interactivity |

## Appendix A

**Table A1.** Item—Construct Correlations.

| Const-ruct | Item | PREP | PFAM | PSQ | PEOU | TRST | WTU | GOV | EXPL | LANG |
|---|---|---|---|---|---|---|---|---|---|---|
| PREP | PREP2 | 0.6989 (8.3966 **) | | | | | | | | |
| | PREP3 | 0.8931 (57.995 **) | | | | | | | | |
| PFAM | PFAM1 | | 0.8289 (42.0390 **) | | | | | | | |
| | PFAM4 | | 0.7509 (18.9136 **) | | | | | | | |
| PSQ | PSQ1 | | | 0.8178 (41.0290 **) | | | | | | |
| | PSQ2 | | | 0.7237 (17.5371 **) | | | | | | |
| PEOU | PEOU 1 | | | | 0.8215 (28.7253 **) | | | | | |
| | PEOU 2 | | | | 0.8538 (27.1163 **) | | | | | |
| | PEOU 3 | | | | 0.8541 (29.2614 **) | | | | | |
| TRST | TRST1 | | | | | 0.8439 (36.6826 **) | | | | |
| | TRST2 | | | | | 0.8742 (41.1002 **) | | | | |

**Table A1.** *Cont.*

| Const-ruct | Item | PREP | PFAM | PSQ | PEOU | TRST | WTU | GOV | EXPL | LANG |
|---|---|---|---|---|---|---|---|---|---|---|
| WTU | WTU1 | | | | | | 0.7124 (15.298 **) | | | |
| | WTU2 | | | | | | 0.7926 (16.857 **) | | | |
| GOV | GOV1 | | | | | | | 0.8439 (36.6826 **) | | |
| | GOV2 | | | | | | | 0.8742 (41.1002 **) | | |
| EXPL | EXPL1 | | | | | | | | 0.7124 (15.291 **) | |
| | EXPL3 | | | | | | | | 0.7926 (16.853 **) | |
| LANG | LANG1 | | | | | | | | | 0.7124 (15.291 **) |
| | LANG2 | | | | | | | | | 0.7926 (16.853 **) |

Loading (T-Value **) ** Indicates *p*-value < 0.01.

## Appendix B

**Table A2.** Correlation Coefficient between Constructs.

| Constructs | | PREP | PFAM | PSQ | PEOU | TRST | WTU | GOV | EXPL | LANG |
|---|---|---|---|---|---|---|---|---|---|---|
| PREP | Pearson Correlation | 1.000 | −0.035 | −0.007 | −0.040 | 0.078 | 0.116 * | 0.059 | 0.191 ** | 0.189 ** |
| | Sig. (2-tailed) | 0.0 | 0.546 | 0.898 | 0.495 | 0.179 | 0.046 | 0.314 | 0.002 | 0.013 |
| PFAM | Pearson Correlation | −0.035 | 1.000 | 0.553 ** | 0.402 ** | 0.323 ** | 0.319 ** | 0.127 * | 0.058 | 0.067 |
| | Sig. (2-tailed) | 0.556 | 0.0 | 0.000 | 0.000 | 0.000 | 0.000 | 0.029 | 0.339 | 0.348 |
| PSQ | Pearson Correlation | −0.007 | 0.553 ** | 1.000 | 0.469 ** | 0.244 ** | 0.198 ** | 0.039 | 0.114 | 0.118 |
| | Sig. (2-tailed) | 0.898 | 0.000 | 0.0 | 0.000 | 0.000 | 0.000 | 0.500 | 0.057 | 0.067 |
| PEOU | Pearson Correlation | −0.040 | 0.402 * | 0.479 ** | 1.000 | 0.492 ** | 0.515 ** | 0.160 ** | 0.133 * | 0.156 * |
| | Sig. (2-tailed) | 0.495 | 0.000 | 0.000 | 0.0 | 0.000 | 0.000 | 0.005 | 0.027 | 0.058 |
| TRST | Pearson Correlation | 0.078 | 0.323 ** | 0.244 ** | 0.492 ** | 1.000 | 0.557 ** | 0.083 | 0.090 | 0.072 |
| | Sig. (2-tailed) | 0.179 | 0.000 | 0.000 | 0.000 | 0.0 | 0.000 | 0.149 | 0.134 | 0.125 |
| WTU | Pearson Correlation | 0.116 * | 0.319 ** | 0.198 ** | 0.515 ** | 0.557 ** | 1.000 | 0.118 * | 0.090 | 0.081 |
| | Sig. (2-tailed) | 0.046 | 0.000 | 0.000 | 0.000 | 0.000 | 0.0 | 0.040 | 0.133 | 0.144 |
| GOV | Pearson Correlation | 0.059 | 0.127 * | 0.039 | 0.160 ** | 0.083 | 0.118 * | 1.000 | 0.128 * | 0.117 * |
| | Sig. (2-tailed) | 0.314 | 0.029 | 0.500 | 0.005 | 0.149 | 0.040 | 0.0 | 0.033 | 0.022 |
| EXPL | Pearson Correlation | 0.191 ** | 0.058 | 0.114 | 0.133 * | 0.090 | 0.090 | 0.128 * | 1.000 | 0.138 * |
| | Sig. (2-tailed) | 0.002 | 0.339 | 0.057 | 0.027 | 0.134 | 0.133 | 0.033 | 0.0 | 0.023 |
| LANG | Pearson Correlation | 0.189 ** | 0.067 | 0.118 | 0.156 * | 0.072 | 0.081 | 0.117 * | 0.138 * | 1.000 |
| | Sig. (2-tailed) | 0.013 | 0.348 | 0.067 | 0.058 | 0.125 | 0.144 | 0.022 | 0.023 | 0.0 |

** Correlation is significant at the 0.01 level (2-tailed). * Correlation is significant at the 0.05 level (2-tailed).

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
