# Peer review of "Factors Affecting mHealth Technology Adoption in Developing Countries: The Case of Egypt"

_computers, doi:10.3390/computers12010009_

Round 1

Reviewer 1 Report

study aims to contribute to the mHealth research strand by looking to understand the spectrum of factors that impact adoption of mHealth interventions by users in the developing world. 

Motivation is not clear 

 update the related work with more recent works such as Personal digital twin: a close look into the present and a step towards the future of personalised healthcare industry, Blockchain-based digital twins collaboration for smart pandemic alerting: decentralized COVID-19 pandemic alerting use case,Blockchain-Empowered Multi-Robot Collaboration to Fight COVID-19 and Future Pandemics

discussion section need more explanation 

the article organisation has an issue such as the conclusion section is missing 

tables should be organised within the margin of the paper 

Author Response

Author’s respond: I would like to thank the reviewers for their valuable recommendations and suggestions, which contributed to the improvement of the paper.

Reviewer#1

Reviewer#1 comment: Extensive English Review

Author’s respond: The paper was reviewed by a language editor

Reviewer#1 comment: The study aims to contribute to the mHealth research strand by looking to understand the spectrum of factors that impact adoption of mHealth interventions by users in the developing world. 

Motivation is not clear 

Author’s respond: I modified the research aim consistently overall the paper to be “The study aims to provide implications for mHealth designers and service providers on how to promote the adoption of their applications in developing countries”

Reviewer#1 comment: Improve the introduction to provide sufficient background and to include all relevant references

Author’s respond: I added the following paragraph in the introduction section:

“mHealth apps are generally designed to provide healthcare information to the public, collect health data, monitor patients remotely, access medical records, conduct diagnoses, and assist in disease prevention and management (Vaghefi, et al., 2019). They are classified as education and teaching apps for health care workers, health and fitness apps for patients and the public, remote symptom-tracking apps, and others (Ryu, 2012). The COVID-19 pandemic has led to the development of information apps, providing the public with information on the health measures to be taken and to suggest medical service providers (Singh et al., 2020). 

Despite the vast evidence in the literature highlighting the benefits of mHealth apps, poor public adoption remains a challenge (Jembai et al., 2022). Different patterns of adoption were found among different demographics, where socioeconomic factors were found to be one barrier to the adoption (Peprah et al., 2019). Hence, more studies on mHealth apps adoption are needed, especially for citizens from un-explored culture. This study aims to provide implications for mHealth designers and service providers on how to promote the adoption of their applications among citizens in developing countries “

Reviewer#1 comment: update the related work with more recent works such as Personal digital twin: a close look into the present and a step towards the future of personalized healthcare industry, Blockchain-based digital twins collaboration for smart pandemic alerting: decentralized COVID-19 pandemic alerting use case, Blockchain-Empowered Multi-Robot Collaboration to Fight COVID-19 and Future Pandemics

Author’s respond: Thank you for this suggestion, I added the following paragraph in the Literature Review section:

Future Role of Emerging Technologies in Health Services

Some recent research (Alsamhi & Lee, 2021; Alsamhi et al., 2022) highlighted the possible future potential of blockchain-based multi-robot collaboration in health care environment. Such studies provide recommendations for academic and industrial researchers to give more effort into the blockchain, multi-robot collaboration, and smart IoT technologies towards smart health care environment.

The work of Alsamhi et al. (2022) highlighted the future possible impact of digital twins, blockchain, Internet of Things (IoT), and Artificial Intelligence (AI) in driving a revolution in the healthcare sector. They suggested a blockchain-based collaborative digital twins framework for decentralized epidemic alerting to combat COVID-19 and any future pandemics. The suggested framework has the potential to decentralize epidemic alerting to combat diseases outbreaks, and to facilitate secure real-time data exchange and analysis across multiple participants in the time pf pandemics.

The work of Alsamhi & Lee (2021) highlighted the challenges and opportunities of integrating blockchain and multi-robot to combat pandemics. They proposed a framework to increase the intelligence, decentralization, and autonomous operations of connected multi-robot collaboration in the blockchain network. They proposed specific applications for Blockchain-Empowered Multi-Robot, such as integration between smart IoT devices and multi-robot in a smart hospital environment.

Reviewer#1 comment: discussion section need more explanation 

Author’s respond: Thank you for this valuable recommendation, the following paragraph was added in the discussion section:

Some of these reported themes are suggested, in other technology adoption context such as e-commerce, to reflect local contextual or cultural factors. Some studies argue that Hofstede’s UA-uncertainty avoidance (Hofstede, 2001) culture variable plays an important role in the attitude forming towards technology adoption in developing countries (Kortemann, 2005), and namely in the Arab countries (Shoib et al., 2003). These studies argued that people in a developing culture and with a high score for UA may not gratuitously accept new technologies, especially these which are directly related to their lives. Technology users with high uncertainty avoidance are expected to have a low tolerance for ambiguity and uncertainty; while using mHealth apps can be seen as an example of an activity with an uncertain outcome (Peprah et al., 2019). More specifically, previous research suggested that for users from uncertainty avoidance culture, such as the Egyptians, the effect of perceived reputation (El said et al., 2009), perceived familiarity (Lee et al., 2017), perceived risk (Ko et al., 2004), and trust (Gefen, 2003) on technology adoption are significant. 

Receiving health advice from a mobile application presents numerous risks for users (Peprah et al., 2019). The process becomes less uncertain when the app provider is one with a good reputation and trustworthy; uncertainty might also decrease when the user is familiar with this app. It is therefore tempting to explain the influence of the mHealth app perceived reputation, familiarity, risk, and trust on adoption within the sample of the current research.  While these factors might be also important for low uncertainty avoidance cultures such as the US and Australia, Nevertheless, it is suggested that the effect of these factors on adoption of mHealth apps to be relatively more important in high uncertainty avoidance cultures”.

Reviewer 2 Report

Dear Authors.

Thanks for submitting this work. The paper provides a method for mHealth application usage in Egypt The paper is good but it has some minor comments.

Points in favor

+ Good statistical analysis

Points against:

- missing conclusion section.

-limited discussion.

Detailed feedback:

The abstract looks lengthy, kindly reduce the number of words to a maximum of 250 words or as per journal requirements.

Introduction

please clarify the research problem. Are we discussing the technologies usage, thereby TAM related studies must be framed out. are we discussing usage, so that quality in use must be addressed?

references include these

https://ieeexplore.ieee.org/abstract/document/9558838

https://www.sciencedirect.com/science/article/abs/pii/S0950584922001422

Methodology

it is short, we should see why semi-structured interviews were used. how we chose the sample, and how it applies to the research problem

Where Research Findings are available, please discuss what does the findings imply a separate section for discussion is empirical.

Kindly separate limitations in a separate section.

place the conclusion in a separate section. The current one is very short and does not imply a conclusion

Author Response

Reviewer#2 comment: 

The abstract looks lengthy, kindly reduce the number of words to a maximum of 250 words or as per journal requirements.

Author’s respond: The abstract was reduced to 250 words, as following:

Mobile health apps are seeing rapid growth with the potential to improve access to health care services for disadvantage communities, while enhancing the efficiency of healthcare delivery value chain. Still, the adoption of mHealth apps is relatively low, especially in developing countries. In Egypt, an initiative for national-level health care coverage was launched in 2021, accompanied with a rise of mHealth start-ups. However, many of these projects did not progress beyond the pilot stage, with very little known about antecedents of mHealth adoption for the Egyptian user. Semi-structure interviews were conducted with 22 Egyptians, aiming to uncover factors affecting the use of mHealth apps for Egyptian citizens. Some of these factors were introduced by previous studies, such as: Perceived Service Quality, Perceived Risk, Perceived Ease of Use, and Trust, others were not well established in mHealth research strand, such as: Perceived Reputation, and Perceived Familiarity, while Governance, Personalized Experience, Explain-ability, Interaction, Language, and Cultural Issues, are novel factors introduced by the current research. The effect of these suggested independent variables on the willingness to adopt mHealth apps was validated using a survey administrated with 150 Egyptians, confirming the significant positive effect of most of these factors on the mHealth adoption in Egypt. This research contributes to methodology by introducing novel constructs in the mHealth research context, which might be specific for the target developing country. Practical implications were suggested for designers and health care service providers might increase the adopting of their apps in developing countries, such as Egypt.

Reviewer#2 comment: 

Introduction: please clarify the research problem. Are we discussing the technologies usage, thereby TAM related studies must be framed out. are we discussing usage, so that quality in use must be addressed?

Author’s respond: Thank you for this valuable comment. The paper is about adoption. I extended the literature review by adding adoption research, and added the following paragraph in the literature review section

“While various models investigated challenges to systems’ adoption in general  (Obaidi  et al., 2022; Atoum et al., 2021), little work focused on mobile health adoption challenges in specific. Murnane et al. (2015) studied popular mobile health and fitness applications to examine reasons underlying both app adoption and abandonment of these applications. The study explored reasons why citizens download health apps in the first place, what motivated continued usage, and what reasons prompted app abandonment. It was reported that the top reasons for maintaining use of a health mobile app were perceived effectiveness, efficacy, and customization ability. Reasons for app abandonment, on the other hand, were reported as miss-trust of service provider, lack of desired features such as tracking and notifications, and lack of motivation.

Peprah et al. (2019) looked at mHealth apps adoption among students in Ghana. The study observed gender inequalities in the use of mHealth apps, where adoption of males was higher.  Other demographic and socioeconomic factors were reported to play a major role in mHealth adoption decisions, such as socio-demographic characteristics, specifically ethnicity, class of respondents and high average monthly income. The study suggested that these could be as sociocultural barriers to potential mHealth apps innovation adoption across citizens in Ghana”.

Reviewer#2 comment: 

references include these

https://ieeexplore.ieee.org/abstract/document/9558838

https://www.sciencedirect.com/science/article/abs/pii/S0950584922001422

Author’s respond: Both references were included in the literature review and refrences sections

Reviewer#2 comment: 

Where Research Findings are available, please discuss what does the findings imply a separate section for discussion is empirical.

Author’s respond: A Discussion section was added the following paragraph was added in the discussion section:

Some of these reported themes are suggested, in other technology adoption context such as e-commerce, to reflect local contextual or cultural factors. Some studies argue that Hofstede’s UA-uncertainty avoidance (Hofstede, 2001) culture variable plays an important role in the attitude forming towards technology adoption in developing countries (Kortemann, 2005), and namely in the Arab countries (Shoib et al., 2003). These studies argued that people in a developing culture and with a high score for UA may not gratuitously accept new technologies, especially these which are directly related to their lives. Technology users with high uncertainty avoidance are expected to have a low tolerance for ambiguity and uncertainty; while using mHealth apps can be seen as an example of an activity with an uncertain outcome (Peprah et al., 2019). More specifically, previous research suggested that for users from uncertainty avoidance culture, such as the Egyptians, the effect of perceived reputation (El said et al., 2009), perceived familiarity (Lee et al., 2017), perceived risk (Ko et al., 2004), and trust (Gefen, 2003) on technology adoption are significant. 

Receiving health advice from a mobile application presents numerous risks for users (Peprah et al., 2019). The process becomes less uncertain when the app provider is one with a good reputation and trustworthy; uncertainty might also decrease when the user is familiar with this app. It is therefore tempting to explain the influence of the mHealth app perceived reputation, familiarity, risk, and trust on adoption within the sample of the current research.  While these factors might be also important for low uncertainty avoidance cultures such as the US and Australia, Nevertheless, it is suggested that the effect of these factors on adoption of mHealth apps to be relatively more important in high uncertainty avoidance cultures”.

Reviewer#2 comment: Justification for sampling technique and

Limitation section

Author’s respond: Justification and limitation of using convenient sampling is clarified by adding the following paragraph in the limitation and future work section

Convenience sampling was used in the current research for its cost and time saving, also because it is a simple technique that allows researcher to obtain basic data and trends regarding the phenomena under investigation without the complications of using a randomized sample. Still, the sample could be a limitation in the current research. As most of the sample were familiar with mobile technology, with good English language, this might hinder a possible socio-cultural effect on adoption.  Future studies should incorporate participants from rural areas, with various demographic characteristics to assess regional differences and possible cultural dimensions.

Round 2

Reviewer 1 Report

The authors addressed my comments very well. only references cited in text should be appear in the section references  

Reviewer 2 Report

Thanks for the edits.